# Flexible Pressure Sensors Based on P(VDF-TrFE) Films Incorporated with Ag@PDA@PZT Particles

**DOI:** 10.3390/s24165415

**Published:** 2024-08-21

**Authors:** Yingzheng Mei, Chuan Cao, Peng Zhou, Jianqiao Wang, Miaoxuan Liu, Xunzhong Shang, Juan Jiang, Yajun Qi, Tianjin Zhang

**Affiliations:** Ministry of Education Key Laboratory for Green Preparation and Application of Functional Materials, Hubei Provincial Key Laboratory of Polymers, Collaborative Innovation Center for Advanced Organic Chemical Materials Co-Constructed by the Province and Ministry, School of Materials Science and Engineering, Hubei University, Wuhan 430062, China

**Keywords:** P(VDF-TrFE), piezoelectric properties, pressure sensors

## Abstract

Films of piezoelectric poly(vinylidene fluoride) (PVDF) and its copolymer P(VDF-TrFE) have been studied intensively for their potential application in piezoelectric sensing devices. The present work focuses on tuning the piezoelectric properties of P(VDF-TrFE) films via incorporating Ag and polydopamine co-decorated PZT (Ag@PDA@PZT) particles. Ag@PDA@PZT particles can effectively improve the β-phase content and piezoelectric properties of P(VDF-TrFE) films. The highest open-circuit voltage and short-circuit current of P(VDF-TrFE)-based flexible pressure sensors incorporating Ag@PDA@PZT particles are ~30 V and ~2.4 μA, respectively. The flexible sensors exhibit a response to different body movements, providing a practical and potentially useful basis for human−machine interface applications.

## 1. Introduction

Self-powered flexible pressure sensors that involve the use of organic/inorganic functional materials have drawn more and more attention due to their potential applications in the fields of human physiological information monitoring, robot tactile perception, and human–computer interaction [1,2,3,4,5,6]. Poly(vinylidene fluoride) (PVDF) and its copolymer poly(vinylidene fluoride-co-trifluoroethylene) (P(VDF-TrFE)) are the key flexible polymers that have piezoelectric properties due to their mechanical and thermal stabilities [7,8]. These piezoelectric polymers can be widely utilized in the field of pressure sensing. Specifically, P(VDF-TrFE) has five different polymorphs, i.e., α, β, γ, δ, and ε phases, of which the β phase exhibits the highest piezoelectric response. Therefore, increasing the content of the β phase is of great importance for improving the performance of P(VDF-TrFE)-based devices.

Incorporating inorganic piezoelectric ceramic particles into P(VDF-TrFE) is one of the effective methods of improving the piezoelectric properties of P(VDF-TrFE)-based composites. Zhou et al. reported that a piezoelectric enhancer of dimethoxymethylvinylsilane-modified BaTiO_3_ (M-BTO) could improve the dispersion of ceramic particles and the piezoelectricity of M-BTO/PVDF composites. The output voltage and current of the composite reached 17.2 V and 421.1 nA, respectively [9]. In order to tune the dielectric constants, breakdown strength, and dielectric loss, a TiO_2_ shell was introduced onto the barium strontium titanite (BST) surface. The results showed that the TiO_2_-decorated BST/PVDF nanocomposites exhibited enhanced dielectric constants, higher breakdown strength, and suppressed dielectric loss, which were related to the inter-particle and intra-particle polarization and the mitigation of the interface mismatch [10]. Another interesting method of improving piezoelectricity was constructing a morphotropic-phase boundary in P(VDF-TrFE), as the interactions between the ZnO fillers and the -CH_2_, -CF_2_, and -CHF groups in P(VDF-TrFE) were conducive to enhancing the piezoelectric responses in the phase transition region [11]. SnO_2_ has also been demonstrated to be an appropriate filler that could increase the piezoelectric output performance via strengthening the local electric field [12]. So far, we have seen that inorganic ceramic fillers, including SrTiO_3_ [13], barium titanate [14], ZnSnO_3_ [15], and lead zirconate titanate (PZT), can significantly improve the piezoelectric properties of P(VDF-TrFE).

Increasing the dispersion and decreasing the agglomeration of piezoelectric ceramic fillers in the P(VDF-TrFE) matrix contribute to a higher piezoelectric response, as more fillers can induce a higher polarization degree. The degree of spontaneous polarization reorientation of piezoelectric ceramic fillers under a poling electric field is low due to the high resistance of the P(VDF-TrFE) matrix [16]. A relative increase in the conduction of P(VDF-TrFE)-based piezoelectric composites can promote the partial voltage applied to the ceramic fillers, which is conducive to the enhancement in both the degree of spontaneous polarization reorientation of the ceramic fillers and the piezoelectric response of composites [17,18]. By designing core–shell-structured inorganic fibers and sandwich-structured films, Cui et al. demonstrated excellent energy storage density and efficiency in blended polymer-based composites [19]. Ag-decorated potassium–sodium niobite particle–polymer composites generated the high open-circuit voltage and short-circuit current of 282 V and 32.2 μA, respectively [20]. Similarly, Ag-decorated polydopamine-functioned BaTiO_3_/PVDF composites exhibited enhanced piezoelectricity compared to composites not decorated with Ag [21]. Sasmal et al. demonstrated that space-charge-induced dielectric permittivity and energy harvesting ability could be improved by incorporating nano-Ag-decorated ZnSnO_3_ into PVDF-based composites [22]. In addition, some other fillers, including D-phenylalanine chelate with ZnO nanoparticles [23], Nb_2_CT_x_ and BaTiO_3_ [24,25], Sb nanosheets and BaTiO_3_ [26], TiO_2_@Fe_3_O_4_@ethylenediamine hybrid nanowires [27], carbon nanotubes [28,29,30,31], and reduced graphene oxide nanosheet [32], have been proven effective in eliminating the agglomeration of fillers and increasing the conduction of composites.

Lead zirconate titanate (PZT) ceramics exhibit an ultra-high piezoelectric coefficient. So far, the dispersion of PZT in piezoelectric polymers and the enhancement in the piezoelectric response of P(VDF-TrFE)-based polymers incorporating PZT have not been well studied. Polydopamine (PDA) facilitates the homogeneous dispersion of PZT in P(VDF-TrFE) [33]. PDA also acts as a binder between PZT and P(VDF-TrFE), which gives rise to a stronger self-polarization effect. Self-polarization can increase the polarity of P(VDF-TrFE), leading to an increase in the content of the β phase [34]. Silver (Ag) nanoparticles can build an electrical connection with PZT, which is conducive to the increase in the polarization of piezoelectric composites [16].

In this work, based on our previous study [34], PZT-incorporated P(VDF-TrFE) films were fabricated, and their piezoelectric properties were studied in detail, where PZT was covered with polydopamine (PDA), and silver (Ag) was decorated on the surface of PDA. The relationship between the content of Ag@PDA@PZT particles and the piezoelectric response was established. The results of the current study will be of significance for the application of piezoelectric composites in flexible pressure sensing.

## 2. Materials and Methods

PZT particles were prepared using the conventional solid-state reaction technique [35]. P(VDF-TrFE) (80/20 wt%) powder was purchased from Solvay (Alorton, IL, USA). N,N-dimethyl formamide (DMF) was obtained from Sigma-Aldrich (St. Louis, MO, USA). Dopamine hydrochloride (C_8_H_12_ClNO_2_, 98%) was purchased from Macklin (Shanghai, China). Tris buffer solution (pH = 8.5) was obtained from Phygene Co., Ltd. (Fuzhou, China). Silver nitrate solution was purchased from Howei Pharm Co., Ltd. (Guangzhou, China).

PZT (0.5 g) and dopamine hydrochloride (0.5 g) were added to Tris buffer solution (40 mL) and stirred for 24 h. After that, the solution was centrifuged (at a speed of 3000 rpm for 3 min) and washed repeatedly until the supernatant was transparent. It was then dried at a temperature of 60 °C for 24 h to obtain polydopamine-coated PZT (PDA@PZT).

Silver ammonia solution was obtained by adding ammonia solution (3%) to 40 mL of silver nitrate solution (0.1 mol/L). The silver ammonia solution was then mixed with 0.5 g of PDA@PZT and stirred for 24 h. Ag-decorated PDA@PZT (Ag@PDA@PZT) particles were synthesized after the processes of centrifugating, washing, and drying.

Ag@PDA@PZT particles were added to acetone (2 mL) to form a solution (I). P(VDF-TrFE) solution (II) was obtained by adding 1.5 g of P(VDF-TrFE) powder to 3 mL of DMF and stirring for 2 h. Solutions I and II were mixed and stirred for 12 h to form solution (III). Ag@PDA@PZT/P(VDF-TrFE) piezoelectric composites were fabricated by coating solution III on a glass slide and spin-coated at 1000 rpm for 30 s, followed by a quick transfer and heating at 60 °C for 30 min (see Appendix A for detailed preparation process). Different contents of Ag@PDA@PZT in the composites (including 0 wt%, 3 wt%, 6 wt%, 10 wt%, and 20 wt%) were realized by adding 0 g, 0.03 g, 0.06 g, 0.1 g, or 0.2 g of Ag@PDA@PZT particles to solution I. The flexible pressure sensors were obtained by attaching aluminum foil as an electrode on both sides of the piezoelectric composites, which were then sealed using polyimide tape (see Appendix A for schematic diagram of sample structure).

Morphology of the film surface was characterized by using field-emission scanning electron microscopy (FESEM, Sigma 500, Carl, Oberkochen, Germany). The morphology of a single particle was characterized using transmission electron microscopy (TEM, JEM-F200, JOEL, New York, NY, USA). The microscopic structure of the composites was examined using X-ray diffraction (XRD, D8 Advance, Bruker, Billerica, MA, USA) and Fourier-transform infrared spectroscopy (FTIR, Nicllet iS50, Thermo Fisher Scientific, Waltham, MA, USA). The local piezoelectric properties of the films were explored using piezoresponse force microscopy (PFM, MFP-3D Origin, Asylum Research, Santa Barbara, CA, USA). Compositional analyses of the composites were conducted using X-ray photoelectron spectroscopy (XPS, Escalab 250Xi, Thermo Fisher, Waltham, MA, USA). The dynamic pressure applied to the piezoelectric composites was measured via a digital force gauge (Aipu Metrology Instrument Co., Ltd., Guangzhou, China). Short-circuit current and output voltage were measured using an electrometer (Keithley 2450, Solon, OH, USA) and oscilloscope (TBS1072B, Tektronix, Beaverton, OR, USA), respectively.

## 3. Results

Figure 1a shows the morphology of a single PZT particle covered with PDA, where the inner black part and outer gray part represent PZT and PDA, respectively. PDA covers most of the surface of PZT, demonstrating the successful fabrication of a PDA@PZT particle. A single PDA@PZT particle covered with Ag nanoparticles is shown in Figure 1b; we can see some small nanoparticles with a size of ~20 nm on its surface. Since Ag nanoparticles exhibit a similar color to that of PDA, XPS measurement was conducted to further verify the existence of Ag on the surface of PAD@PZT particles. Figure 1c,d show the XPS spectra of O 1s for PAD@PZT (before Ag was covered) and Ag@PAD@PZT (after Ag was covered). Both spectra exhibit the same binding energy of O 1s (~532 eV), as well as fitted binding energies that correspond to -C=O (~531.4 eV) and -C-OH (~532.8 eV), revealing that Ag does not change the chemical state of O 1s of the particles. On the other hand, however, the related contents of -C=O and -C-OH change with the incorporation of Ag nanoparticles, as illustrated in Figure 1e. The relative content of -C-OH is higher in PDA@PZT particles, while the relative content of -C=O is higher in Ag@PDA@PZT. Hence, incorporating Ag can affect the contents of both -C-OH and -C=O in PAD@PZT particles, indicating that Ag had been covered on the surface of PAD@PZT particles. In addition, Ag could be detected in PDA@PZT particles, as seen in the XPS spectrum in Figure 1f, which confirms that Ag@PDA@PZT particles had been synthesized. The TEM-EDS mappings of the Ag@PDA@PZT particle and PDA@PZT particle shown in Appendix A, respectively, also prove the successful synthesis of Ag@PDA@PZT particles.

Figure 2a shows the morphology of a P(VDF-TrFE) film containing Ag@PDA@PZT particles; it is dense and without obvious holes or defects. The microstructures of PZT and Ag@PDA@PZT were measured via XRD, as shown in Figure 2b. Both XRD patterns exhibit the same diffraction peaks, revealing that the process of covering PDA and Ag on PZT particles does not affect the microstructure of PZT. It is expected that incorporating Ag@PDA@PZT particles in P(VDF-TrFE) effectively increases the dispersion of PZT in P(VDF-TrFE) as well as the conduction of P(VDF-TrFE) films [20,36].

To verify the effect of Ag and PDA on the piezoelectric properties of P(VDF-TrFE) films, COMSOL Multiphysics simulation (refer to the Appendix A for detailed simulation parameters) was used to simulate the potential distribution under the same applied force, as illustrated in Figure 3. Different colors represent the intensity of the potential, and the starting potential was set to 0 V. The output potential of the PDA@PZT-incorporated P(VDF-TrFE) film was increased significantly compared with that of the merely PZT-incorporated film, as seen in Figure 3a,b. Similarly, by incorporating Ag@PDA@PZT particles, the output potential of the P(VDF-TrFE) film was as high as 118 V, which is obviously higher than that achieved by incorporating PDA@PZT particles. Figure 3d shows the trend in the simulated piezoelectric potential when incorporating different fillers, confirming that Ag and PDA co-decorated PZT particles can effectively improve the piezoelectric response of P(VDF-TrFE) film.

There is a close relationship between the content of Ag@PDA@PZT particles and the piezoelectric response of P(VDF-TrFE) films. The piezoelectric response of the films was characterized using PFM, as shown in Figure 4, where the content of the Ag@PDA@PZT particles was 0 wt%, 3 wt%, 6 wt%, 10 wt%, and 20 wt%. The variation in the average amplitude in Figure 4a–e with the content of the Ag@PDA@PZT particles is summarized in Figure 4f. With the increase in Ag@PDA@PZT content, we first see an increase in the piezoelectric response of the films. At a Ag@PDA@PZT content of 10 wt%, the average amplitude (piezoelectric response) reaches the maximum value of 175 pm. A further increase in the Ag@PDA@PZT content gives rise to a decline in the piezoelectric response. The amplitude and phase curves measured at a specific point on the film with a Ag@PDA@PZT content of 10 wt% are shown in Figure 4g,h; they also exhibit good piezoelectric properties.

The content of the β phase was measured via XRD and FTIR. Figure 5a presents the XRD patterns of the P(VDF-TrFE) films with different contents of Ag@PDA@PZT particles. The (020) lattice plane of the α-phase P(VDF-TrFE) and the (110/200) lattice plane of the β-phase P(VDF-TrFE) are observed around 2θ values of 16.5° and 19.5°, respectively [23,34]. Other diffraction peaks are all from PZT, which is inconsistent with the XRD result in Figure 2b. The β-phase content can be determined from the ratios of diffraction peak intensity at 2θ values of 16.5° and 19.5° (I_(19.5)_/I_(16.5)_), where a higher value of I_(19.5)_/I_(16.5)_ corresponds to a higher β-phase content [25,37]. Figure 5d shows the variation in I_(19.5)_/I_(16.5)_ with Ag@PDA@PZT content. At around 10 wt%, the P(VDF-TrFE) film exhibits the highest I_(19.5)_/I_(16.5)_ value. Hence, the composite film with 10 wt% Ag@PDA@PZT particles has the highest β-phase content. FTIR is another method that can be used to explore the β-phase content. Figure 5b shows the FTIR spectra of the P(VDF-TrFE) films with different contents of Ag@PDA@PZT particles. The absorption peaks corresponding to the α phase and β phase are located at around 763 cm^−1^ and 840 cm^−1^, respectively. The relative content of the β phase in the composites can be calculated based on the following formula [38,39]:$$F\left(\beta \right)=\frac{A_{\beta }}{\left(\frac{K_{\beta }}{K_{\alpha }}\right)A_{\beta }+A_{\alpha }}$$
where *A*_α_ and *A*_β_ are the absorbances at 763 cm^−1^ and 840 cm^−1^, respectively, while *K*_α_ and *K*_β_ are constants. Therefore, a higher value of *A*_β_/*A*_α_ gives rise to a higher content of the β-phase. The variation in I_(840)_/I_(763)_ (i.e., *A*_β_/*A*_α_) with the content of Ag@PDA@PZT is illustrated in Figure 5e, which exhibits a trend of first increasing and then decreasing. At around 10 wt%, I_(840)_/I_(763)_ reaches the highest value. Therefore, the content of the β phase is the highest for films with 10 wt% of Ag@PDA@PZT particles, which is inconsistent with the XRD result.

Since the β phase exhibits the highest piezoelectric response, the composite films with various contents of the β phase should exhibit different piezoelectric outputs. Figure 5c,f show the open-circuit voltage and short-circuit current of the flexible pressure sensors assembled based on the P(VDF-TrFE) films with different contents of Ag@PDA@PZT particles, respectively. Both the open-circuit voltage and short-circuit current increase firstly with an increase in the content of Ag@PDA@PZT particles. At a content of 10 wt%, they reach the highest values of ~30 V and ~2.4 μA, respectively. Further increasing the content of Ag@PDA@PZT particles leads to a decrease in the open-circuit voltage and short-circuit current. This trend is the same as that of the β-phase content (see Figure 5d,e), revealing that a higher content of the β phase is conducive to the improvement in the piezoelectric output of flexible pressure sensors.

The relationship between the piezoelectric output of P(VDF-TrFE) films and the content of the Ag@PDA@PZT particles was examined via COMSOL Multiphysics simulation (refer to the Appendix A for detailed simulation parameters). Figure 6a–e show the potential distribution under the same applied force for different contents of Ag@PDA@PZT particles. The variation in the potential is summarized in Figure 6e. It increases firstly with the increase in the content of Ag@PDA@PZT particles and reaches the highest value of 149 V at a Ag@PDA@PZT content of 10 wt%. The potential decreases when the content of Ag@PDA@PZT further increases. Therefore, the change in potential exhibits the same trend as that of the β phase, open-circuit voltage, and short-circuit current.

Finally, P(VDF-TrFE)-based flexible pressure sensors containing 10 wt% Ag@PDA@PZT particles were assembled. These sensors exhibited obvious piezoelectric responses to flicking, tapping, pressing, walking, running, and hammering, as illustrated in Figure 7. The output voltage in most of the situations reached 10 V. The sensitivity of the flexible pressure sensors with 10 wt% Ag@PDA@PZT is shown in Figure 8. At low and high force ranges, the sensitivities of the pressure sensor were 2.51 V/N and 0.466 V/N, respectively.

## 4. Discussion

As there is strong interaction between the -CF_2_- dipole of P(VDF-TrFE) and the -NH_2_- of PDA, the α phase in P(VDF-TrFE) can be stretched into the piezoelectrically active β phase [34]. Due to the high adhesion ability of PDA, PZT particles can effectively be adhered to P(VDF-TrFE) [40]. Hence, PDA can significantly decrease the aggregation of PZT and improve the dispersion of PZT in P(VDF-TrFE). It can also improve the piezoelectric output of P(VDF-TrFE) films compared to that without PDA, as indicated in Figure 3d. In addition, Ag nanoparticles provide more conductive channels for poling electric fields and promote the partial voltage applied to PZT [16]. The piezoelectric output of P(VDF-TrFE) films can be enhanced via the efficient polarization of PZT. Therefore, the piezoelectric response of Ag@PDA@PZT incorporating P(VDF-TrFE) is much higher than that incorporating PDA@PZT, as shown in Figure 3d. However, because of leakage, too many Ag nanoparticles can deteriorate the piezoelectric output. This is the reason why the average amplitude of PFM, β-phase content, open-circuit voltage, and short-circuit current decrease after the content of the Ag@PDA@PZT particles exceeds 10 wt%.

## 5. Conclusions

In summary, Ag and PDA co-decorated PZT particles were synthesized. These particles could effectively improve the β-phase content and piezoelectric properties of P(VDF-TrFE) films. The highest open-circuit voltage and short-circuit current of P(VDF-TrFE)-based flexible pressure sensors incorporating Ag@PDA@PZT particles reached ~30 V and ~2.4 μA, respectively. The improved piezoelectric output is related to the increased dispersion of PZT in P(VDF-TrFE) via PDA as well as to the conductivity of P(VDF-TrFE) being increased via Ag nanoparticles. The assembled P(VDF-TrFE)-based flexible pressure sensors can detect body movements, which is significant for applications in biosensors.

## Figures and Tables

**Figure 1 sensors-24-05415-f001:**
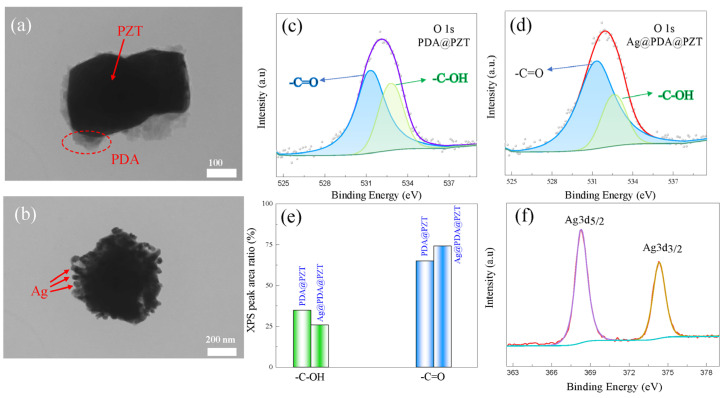
(**a**) TEM image of a PZT particle (black) covered with PDA (gray), (**b**) TEM image of a PZT particle covered with PDA and Ag, where Ag is located on the outer surface of the particle. XPS spectra of O 1s for PDA@PZT (**c**) and Ag@PDA@PZT (**d**), (**e**) XPS peak area ratio of O 1s corresponding to -C-OH and -C=O in PDA@PZT and Ag@PDA@PZT, (**f**) XPS spectrum of Ag3d.

**Figure 2 sensors-24-05415-f002:**
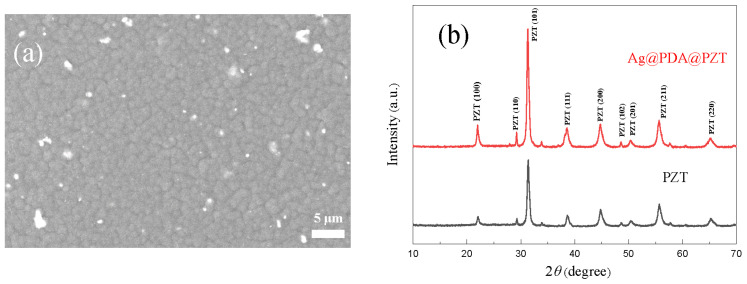
(**a**) SEM image of P(VDF-TrFE) film incorporating Ag@PDA@PZT, (**b**) XRD patterns of PZT powder and Ag@PDA@PZT particles.

**Figure 3 sensors-24-05415-f003:**
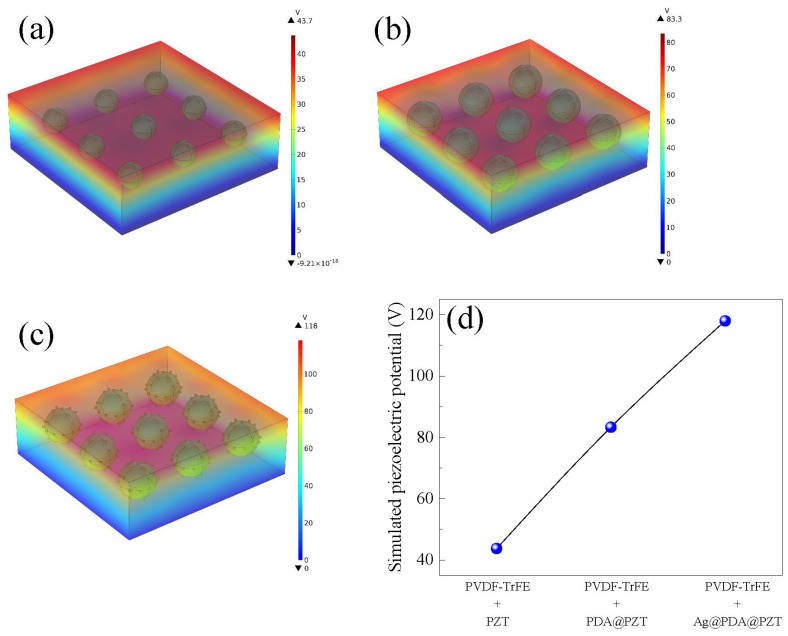
Simulated piezoelectric potential of (**a**) P(VDF-TrFE) incorporating PZT, (**b**) P(VDF-TrFE) incorporating PDA@PZT, (**c**) P(VDF-TrFE) incorporating Ag@PDA@PZT, and (**d**) summary of the simulated piezoelectric potential of (**a**–**c**).

**Figure 4 sensors-24-05415-f004:**
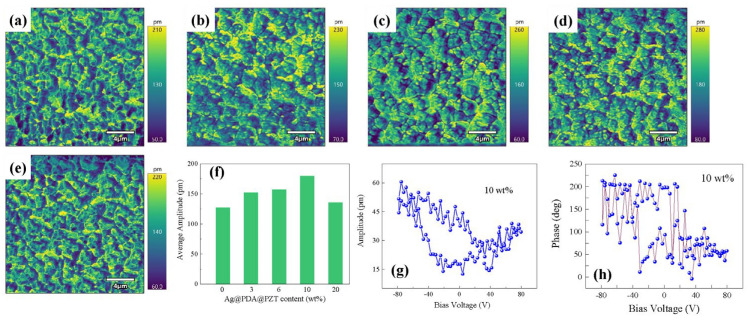
PFM amplitude of P(VDF-TrFE) films with Ag@PDA@PZT contents of 0 wt% (**a**), 3 wt% (**b**), 6 wt% (**c**), 10 wt% (**d**), and 20 wt% (**e**); (**f**) summary of the average amplitude of (**a**–**e**); (**g**) PFM amplitude; and (**h**) PFM phase of P(VDF-TrFE) film with Ag@PDA@PZT content of 10 wt%.

**Figure 5 sensors-24-05415-f005:**
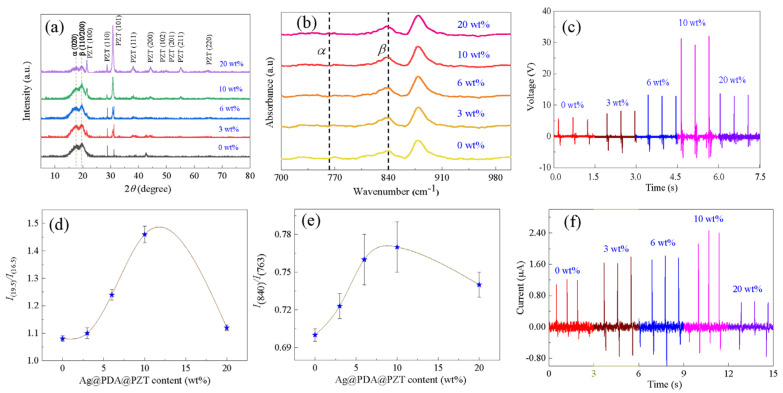
(**a**) XRD patterns of P(VDF-TrFE) films incorporating different contents of Ag@PDA@PZT, where the intensity ratio between peaks located at 2θ of 19.5 and 16.5 is summarized in figure (**d**). (**b**) FTIR spectra of P(VDF-TrFE) films incorporating different contents of Ag@PDA@PZT, where the intensity ratio between peaks located at wave numbers of 763 cm^−1^ and 840 cm^−1^ is summarized in figure (**e**). Open-circuit voltage (**c**) and short-circuit current (**f**) of flexible pressure sensors incorporating different contents of Ag@PDA@PZT.

**Figure 6 sensors-24-05415-f006:**
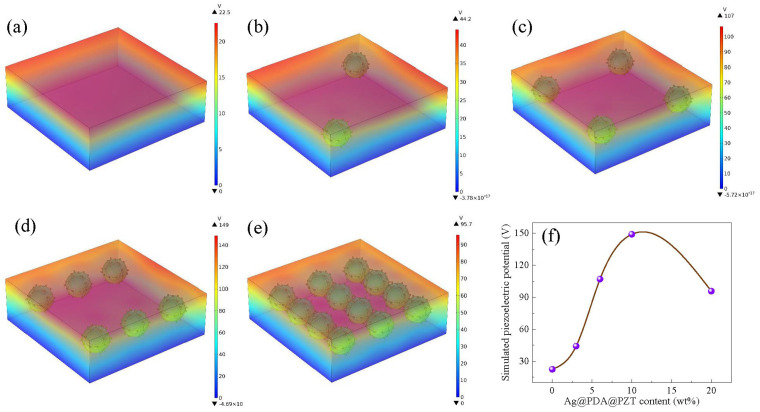
Simulated piezoelectric potential of P(VDF-TrFE) films with Ag@PDA@PZT contents of (**a**) 0 wt%, (**b**) 3 wt%, (**c**) 6 wt%, (**d**) 10 wt%, and (**e**) 20 wt%; (**f**) summary of the simulated piezoelectric potential of P(VDF-TrFE) films with different Ag@PDA@PZT contents.

**Figure 7 sensors-24-05415-f007:**
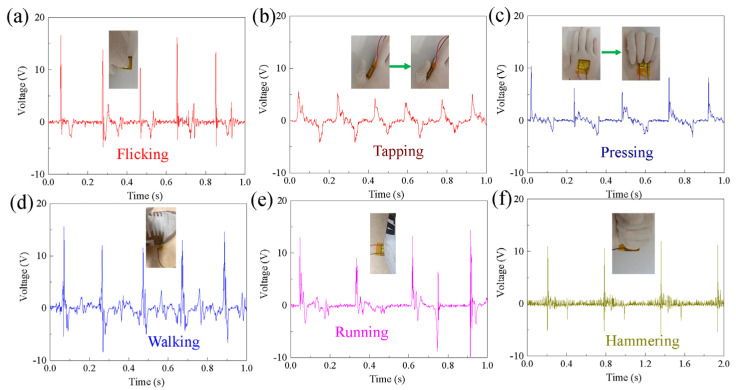
Output voltage of flexible pressure sensors with 10 wt% Ag@PDA@PZT in response to different body movements: (**a**) flicking, (**b**) tapping, (**c**) pressing, (**d**) walking, (**e**) running, (**f**) hammering.

**Figure 8 sensors-24-05415-f008:**
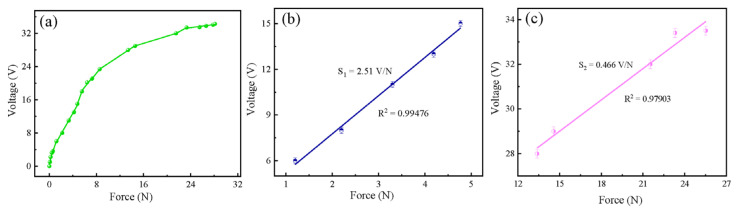
(**a**) Dependence of peak value of output voltage of the pressure sensor on force. Sensitivity of the pressure sensor in the low force range (**b**) and high force range (**c**).

## Data Availability

Data are contained within the article.

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
