# Peer review of "Flexible Pressure Sensors Based on P(VDF-TrFE) Films Incorporated with Ag@PDA@PZT Particles"

_sensors, 2024, doi:10.3390/s24165415_

Round 1

Reviewer 1 Report

Comments and Suggestions for Authors

In this study, the authors investigate the enhancement of piezoelectric properties in P(VDF-TrFE) films by incorporating Ag@PDA@PZT particles. The experimental design is reasonable. However, further improvement is needed, as several points require clarification and some statements need further justification. I recommend publication after major revisions.

1. More experimental details should be provided. For example, the pH value of Tris-DA solution should be provided exactly.

2. The authors are suggested to include TEM-EDS mapping spectra of the prepared particles, especially for PDA@PZT and Ag@PDA@PZT. In addition, the authors should provide a better analysis of the shape of the nanoparticles. SEM images to confirm the size and morphology of the particles (PZT, PDA@PZT and Ag@PDA@PZT) should be presented with the clear analysis.

3. The author claim that "the morphology of P(VDF-TrFE) film containing Ag@PDA@PZT particles, it’s dense and without obvious holes or defects". The cross–sectional morphology of film and corresponding EDS mapping spectra should be included.

4. An explanation should be provided for why the films with 10 wt% of Ag@PDA@PZT particles exhibits the highest β-phase content. Clarifying this aspect will enhance our understanding of how Ag@PDA@PZT affects the material's properties.

5. The statement "Ag nanoparticles not only work as dispersant, but also provide more conductive channels for poling electric field and promote the partial voltage applied to PZT" was presented by the authors but without sufficient elaboration on why Ag nanoparticles can work as dispersant. The authors should give reasonable explanations.

Comments on the Quality of English Language

The English shoud be carefully polised by a native speaker.

Reviewer 2 Report

Comments and Suggestions for Authors

In this paper (sensors-3153876), the authors proposed a pressure sensor based on P(VDF-TrFE) incorporated with Ag@PDA@PZT particlesis. The strategy is useful and results are acceptable, but there are some problems in the writing, introduction, presentation, and discussion. As such, some revisions are needed before possible acceptance. My specific comments are as follows:

1.     Introduction: (1) “human biosignals, energy harvesting, and pressure sensing….”, for “energy harvesting”, it is limited to self-powered sensors. (2) To highlight the advantages of piezoelectric sensors, it is recommended to start the story with various self-powered pressure sensors as the entry point, and may refer to the recent reports such as Sensors 2024, 24(4), 1275; Chem. Eng. J. 2024, 490, 151660. (3) Unclear motivation for material selection. The four materials of P(VDF-TrFE) incorporated with Ag@PDA@PZT particlesis are very complex. What are the functions of each material? How to reasonably determine the components?

2.     The authors should provide a detailed demonstration of the sensor preparation process using a schematic diagram to ensure that the experiment can be repeated.

3.     Morphology characterization: The authors should mark P(VDF-TrFE), Ag, PDA and PZT in the figure.

4.     The authors should provide performance indicators of the sensor, such as sensitivity and detection limit.

5.     It is suggested to summarize the performance parameters of different pressure sensors in a table.

6.     As a research paper, the piezoelectric response mechanism of composites needs systematic analysis.

7.     Simulation: The authors should provide detailed simulation parameters and processes.

8.     The text in some figures is too small, such as Figure 6.

9.     Check the reference format of the journal. For example, due to formatting issues, journal names need to be abbreviated, and numbers in chemical formulas need to be subscripted.

10.  English writing of the manuscript needs to be polished.

Comments on the Quality of English Language

English writing of the manuscript needs to be polished.

Round 2

Reviewer 1 Report

Comments and Suggestions for Authors

The authors have addressed the concerns.

Reviewer 2 Report

Comments and Suggestions for Authors

Concerns of  reviewer  have been considered and addressed properly, and publication is recommended.